# SPARC-RTL: Stochastic Prompt-Assisted RTL Code Synthesis

## Abstract

Large language models (LLMs) show promise in code generation yet struggle with Hardware Description Languages (HDLs) such as Verilog, where models often get stuck in flawed reasoning paths that block accurate synthesis and bug fixing. We introduce Stochastic Prompt Assistance (SPA), a lightweight methodology that leverages LLM prompt sensitivity by injecting controlled, non-semantic perturbations into prompts. This approach encourages exploration of diverse reasoning paths without requiring additional feedback loops. Implemented in the `SPARC-Debugger`, our automated framework pairs SPA with hierarchical error-pattern matching and achieves consistent gains on Verilog code-generation and debugging benchmarks, including critical "0-to-1" unlocks where baseline models fail entirely. SPA thus provides a complementary and orthogonal mechanism to existing decoding and refinement strategies, improving the reliability of LLMs in HDLs and offering a principled path for broader application in high-precision formal reasoning domains.

## 1 Introduction

Hardware Description Languages (HDLs) such as Verilog are foundational for digital circuit design, yet authoring and debugging RTL code remains a notoriously challenging endeavor. The low-level, concurrent semantics of hardware make Verilog verbose and error-prone, leading to time-consuming development cycles and costly iterative debugging Wang et al. (2025b); Rahman et al. (2015). Critically, hardware bugs are immutable post-fabrication, rendering errors exceptionally expensive Makris et al. (2006). These challenges underscore a pressing need for AI-driven assistance in *RTL code synthesis and debugging*.

Large Language Models (LLMs) have demonstrated impressive success in general-purpose code generation (e.g., Google's Gemini 2.5 Pro achieved ∼99% pass@1 on HumanEval PromptLayer Blog (2025); Chen et al. (2021)). However, their effectiveness in the Verilog domain remains limited Thakur et al. (2024); Liu et al. (2023). Verilog's strict syntax, concurrent semantics, and cryptic compiler diagnostics often cause models to *get stuck in flawed reasoning paths*, producing persistent and difficult-to-resolve errors. A notable instance of this phenomenon is the *localization bottleneck*, where models repeatedly misidentify the true source of an error Bui et al. (2022). This tendency is especially problematic in enterprise settings, where privacy constraints necessitate locally hosted models; unfortunately, even the strongest open-source options (e.g., Llama 3.3) exhibit pronounced performance gaps on complex RTL tasks. Together, these factors reveal a *capability gap* that calls for methods which improve reliability under fixed inference budgets.

Concurrently, LLMs exhibit pronounced *prompt sensitivity*: small changes in phrasing can produce dramatic performance swings, with differences of up to 45% observed Cao et al. (2024); Salinas & Morstatter (2024). Rather than treating this sensitivity as a liability, we view it as an opportunity. Our core methodology, **Stochastic Prompt Assistance (SPA)**, leverages controlled, stochastic perturbations of the input prompt to encourage exploration of diverse reasoning trajectories. For example, on the CirFix task `fsm_full`, Llama 3.3 improved from ∼40% success with a fixed prompt to nearly 90% with SPA, and similar $0 \rightarrow 1$ unlocks recur across multiple tasks (Section 5.2). By operating at the input level, SPA provides a lightweight diversification mechanism that is orthogonal to decoding-time strategies such as temperature sampling Holtzman et al. (2020).

To operationalize this methodology, we developed the `SPARC-Debugger`, an automated framework for Verilog code synthesis and debugging that provides both a controlled testbed to isolate SPA's effect and a realistic workflow to demonstrate its practical usefulness. We position SPA not as a monolithic competitor but as a modular, lightweight technique that is complementary to existing state-of-the-art frameworks such as AutoChip Thakur et al. (2023) and MEIC Xu et al. (2024). The name SPARC stands for Stochastic Prompt-Assisted RTL Code Synthesis, reflecting its core refinement loop: when a candidate fails, a hierarchical error-matching mechanism queries dynamic Pattern ($\mathcal{L}_P$) and Error ($\mathcal{L}_E$) Libraries using a tiered matching approach to refine a base prompt ($p_0$). SPA then generates perturbed variants of $p_0$ for resubmission to the model. This cycle of targeted prompt refinement, stochastic diversification, generation, and evaluation progressively *unsticks* the model from brittle reasoning loops, enabling recovery of solutions that baseline prompting cannot reach, thereby improving reliability in RTL debugging and synthesis.

Our primary contributions are: (1) We introduce *Stochastic Prompt Assistance (SPA)*, a lightweight input-level diversification method that leverages prompt sensitivity to expand explored reasoning trajectories and help unstick models from brittle failure modes in RTL code generation and debugging. (2) We design the `SPARC-Debugger`, a practical framework that integrates SPA with hierarchical error-pattern matching inside an iterative refinement loop, intended both as a validation platform for the methodology and as a pluggable component for existing HDL workflows. (3) We evaluate on VerilogEval Liu et al. (2023) (generation) and CirFix Ahmad et al. (2022) (debugging) under fixed decoding settings, observing pass@1 improvements in our setup and several $0 \rightarrow 1$ unlock cases. Taken together, these contributions position SPA as a lightweight and complementary technique for improving the robustness of LLM-based RTL code generation and debugging.

## 2 PRELIMINARY

### 2.1 PROMPT SENSITIVITY AND OUTPUT DIVERSIFICATION

Transformer-based Large Language Models (LLMs) are highly sensitive to input phrasing: even small variations, such as token reordering or punctuation, can shift attention patterns and yield markedly different outputs Vaswani et al. (2017). This "butterfly effect" of prompting has been quantified to produce performance swings of up to 45% between minimally rephrased prompts Cao et al. (2024); Salinas & Morstatter (2024).

To address this variability, prior work has explored two main directions. At the **decoding level**, stochastic strategies such as temperature and nucleus sampling Holtzman et al. (2020) introduce randomness during token generation, while reasoning-oriented approaches like chain-of-thought prompting Wei et al. (2022) and self-consistency Wang et al. (2023) diversify reasoning by sampling multiple trajectories and selecting answers by majority vote. At the **input level**, diversity is created by reformulating prompts. Prompt ensembles Jiang et al. (2023) paraphrase task instructions, DiP-PER Lau et al. (2024) generates varied templates, and multilingual prompting Wang et al. (2025a) leverages cross-lingual variation. More advanced frameworks, including Self-Refine Madaan et al. (2023) and Reflexion Shinn et al. (2023), employ iterative critique or memory-based reflection to improve solutions across multiple rounds. These methods demonstrate the utility of diversification but often require substantial sampling or auxiliary reasoning steps.

Our Stochastic Prompt Assistance (SPA) offers a different perspective. Instead of relying on decoding randomness or complex revision loops, SPA applies lightweight syntactic perturbations at the input level. This approach directly leverages structural sensitivity to broaden reasoning trajectories while keeping inference costs fixed.

### 2.2 CHALLENGES IN LLM-BASED VERILOG CODE SYNTHESIS AND DEBUGGING

Despite advances in general-purpose code generation Chen et al. (2021), Hardware Description Languages (HDLs) such as Verilog remain a challenging domain. RTL tasks demand strict syntactic and semantic precision, while error localization is often hampered by compiler feedback that is cryptic or ambiguous. This difficulty is a well-documented obstacle in program repair and contributes to models becoming stuck in flawed reasoning paths.

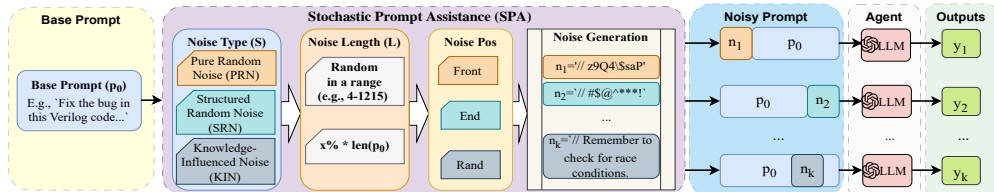

Figure 1: Stochastic Prompt Assistance (SPA). From a base prompt $p_0$, controlled noise injection generates diverse variants, each evaluated by the LLM to expand solution space exploration. The mechanism later integrates into the `SPARC-Debugger` framework.

Existing HDL-specific approaches illustrate these challenges. RTLFixer Tsai et al. (2023) addresses syntax-level errors but struggles with deeper semantic bugs. VeriGen Thakur et al. (2024) and VeriDebug Wang et al. (2024) integrate HDL-specific embeddings and fine-tuning, yet are limited by generalization and context quality. CirFix Ahmad et al. (2022) applies genetic programming to explore candidate patches but scales poorly with larger search spaces. Iterative refinement frameworks such as AutoChip Thakur et al. (2023) and MEIC Xu et al. (2024) leverage compiler feedback to guide repair, achieving higher success on modular tasks but often requiring many iterations and failing when diagnostics are uninformative. Adapting general-purpose agents like FixAgent Lee et al. (2024) to HDL remains difficult due to concurrency and domain-specific semantics.

In summary, prior work reflects two dominant paradigms: tool-based feedback, which depends heavily on diagnostic quality, and diversification strategies, which often incur heavy iterative costs. Both have shown partial success but leave open the need for lightweight mechanisms to expand exploration under ambiguity. This motivates our proposed approach in the next section.

## 3 METHODOLOGY: STOCHASTIC PROMPT ASSISTANCE

Building directly on this motivation, we introduce **Stochastic Prompt Assistance (SPA)** as a distinct input-level ensembling paradigm. Unlike semantic ensembling or multi-round reasoning strategies, SPA employs controlled, low-level syntactic perturbations to "unstick" models from brittle reasoning loops. This design enables exploration of diverse solution paths under fixed inference budgets, which is particularly valuable for domains like Verilog debugging where models frequently get trapped in flawed generative trajectories.

Formally, SPA can be described as a stochastic search process over perturbed prompt variants. Given an initial task description $p_0$, SPA generates an ensemble of $N$ candidate prompts:

$$p_i = p_0 \oplus N_i, \quad N_i \sim \mathcal{D}(S, L, Pos), \quad i = 1, \ldots, N, \tag{1}$$

where $\oplus$ denotes the perturbation operator and $\mathcal{D}(S, L, Pos)$ specifies a distribution over symbol set $S$, noise length $L$, and insertion position $Pos$. Each perturbed prompt $p_i$ is then submitted to the LLM to produce a candidate solution $y_i$. By evaluating this diversified set $\{y_i\}_{i=1}^N$, SPA increases the likelihood that at least one candidate escapes flawed reasoning and aligns with the ground truth, in contrast to $N$ repetitions of the same unperturbed $p_0$.

An overview of this process is shown in Figure 1. The cycle of perturbation, generation, and evaluation constitutes the core of SPA. In the next subsection, we formalize the SPA hypothesis and detail controlled perturbation strategies that underpin this methodology.

### 3.1 THE SPA HYPOTHESIS

The **SPA Hypothesis** is: *Systematic injection of controlled noise into an LLM's input prompt induces meaningful variations in its attention patterns and generative pathways. This controlled perturbation encourages the LLM to explore different input segments (e.g., Verilog code) and consider alternative reasoning paths, such as different error localizations for debugging or varied implementation strategies for code generation. As a result, such injection is hypothesized to increase the probability of finding a successful solution within an ensemble of generated outputs by overcoming single-prompt biases and limitations, without degrading outputs to uncontrolled randomness.* Born from our observations in Verilog debugging, this hypothesis frames controlled noise injection as a principled mechanism for overcoming persistent failure modes in high-precision formal domains.

## 3.2 NOISE TYPES FOR CONTROLLED PERTURBATION

SPA utilizes distinct noise types to perturb prompts, aiming for diverse yet plausible outputs by shifting the LLM's interpretation or focus. These "noises" are alterations designed to achieve this, not arbitrary data corruption. The injection position ($Pos$)—prepending, appending, or random insertion (e.g., as comments)—is also a key parameter for evaluation.

**Pure Random Noise (PRN)** involves injecting tokens or characters with no direct semantic task relevance at various positions ($Pos$). Generated from a broad character set $S$ (e.g., the full ASCII table, including all numbers, letters, and symbols), its length $L$ can be a fixed range (e.g., 4-12 characters like `z9Q4$saP`) or a percentage of the base prompt $p_0$'s length. PRN introduces high local input entropy to maximally perturb nearby contextual embeddings, potentially jolting the model from sub-optimal reasoning. Due to its unpredictable nature, this type of noise is expected to provide a wide search radius, but it also carries the risk of causing topic drift or model confusion, especially if the noise intensity is too high or its placement is inappropriate.

**Structured Random Noise (SRN).** SRN introduces perturbations that, while random, adhere to certain structural or syntactic conventions that the LLM is likely to recognize as ignorable padding or non-instructional content. These noises can be placed at the beginning or end of the prompt, or more commonly, inserted randomly as comments (e.g., `// tmp 854921`, `/* ==== */`) within the prompt, for instance, between code blocks. Other forms include sequences restricted to random digits or random punctuation marks only (from a restricted symbol set $S$, e.g., $S \subset \text{ASCII}_{\text{symbols}}$ or $S \subset \text{ASCII}_{\text{numbers}}$), or the insertion of blank lines at syntactically safe positions. SRN subtly alters token positions and sequence length, thereby changing contextual embeddings and potentially attention patterns, without significantly harming readability or the LLM's interpretation of the core task, as LLMs typically discount such tokens. It aims for moderate diversity with a minimal risk of invalid outputs.

**Knowledge-Influenced Noise (KIN)** injects semantically rich, plausible, in-domain content that is deliberately task-irrelevant or offers a different contextual focus. This content is valid, not erroneous. Examples, inserted at any $Pos$ (often as comments), include common comment sentences from unrelated domains (e.g., `// Know thyself.`), or a short, commented-out Python function for a Verilog task. KIN acts as a sophisticated contextual distractor, compelling the model to discern relevance. It can prime latent LLM knowledge of different coding patterns or syntax, encouraging exploration in a more informed subspace. However, the introduction of out-of-domain semantic content also risks creating a counterproductive cognitive load, and its effectiveness remains an empirical question to be explored.

The distinct noise types (PRN, SRN, KIN) offer varied exploration strategies with inherent trade-offs between scope and stability. This approach of generating diverse prompt variants aligns with established prompt ensembling techniques Li et al. (2022); Jiang et al. (2023), where varied inputs improve model reliability. SPA automates this diversity through controlled noise injection, aiming to increase the likelihood of finding a correct solution within an ensemble, often evaluated by metrics like pass@k Chen et al. (2021).

## 3.3 THEORETICAL BASIS AND CONNECTIONS

The efficacy of Stochastic Prompt Assistance (SPA) is hypothesized to derive from two core mechanisms that connect to broader concepts in LLM research.

### 3.3.1 PERTURBING ATTENTION AND INFERENCE PATHS VIA INPUT NOISE

At a fundamental level, noise injection mechanically impacts the self-attention mechanism of Transformers Vaswani et al. (2017). An input prompt $p_0$, represented by its token embeddings $X$, yields Query ($Q = XW_Q$), Key ($K = XW_K$), and Value ($V = XW_V$) matrices. The attention calculation is $\text{Attention}(Q, K, V) = \text{softmax}\left(\frac{QK^T}{\sqrt{d_k}}\right)V$, where $d_k$ is the dimension of the key vectors. Introducing noise $N_i$ into $p_0$ results in a perturbed embedding sequence $X_i' = \text{Embed}(p_0 \oplus N_i)$. This alteration of $X_i'$ subsequently modifies the Query ($Q_i' = X_i'W_Q$) and Key ($K_i' = X_i'W_K$) matrices, directly changing the attention scores and thus redirecting the model's focus.

In the context of Verilog debugging, we hypothesize that this attentional re-weighting is a crucial mechanism for overcoming persistent failure modes. By altering the attention patterns, SPA can

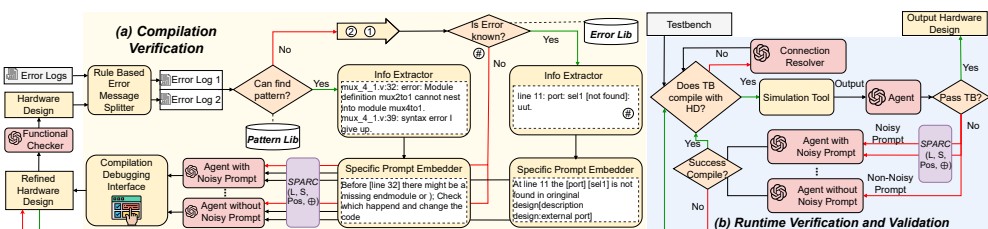

Figure 2: The `SPARC-Debugger` Framework for Verilog Error Diagnosis.

guide the model towards overlooked aspects of the code or error messages. While we do not isolate fine-grained localization accuracy, the consistent $0 \rightarrow 1$ unlocks suggest SPA helps models escape brittle reasoning loops under ambiguous diagnostics (Section 5.2). Each perturbed prompt $p_i$ effectively samples a point near $p_0$ on the input manifold, potentially activating different latent knowledge within the LLM and compelling the model to follow varied inference paths. SPA can thus be conceptualized as a form of Monte Carlo sampling over the prompt space to increase the probability of finding a successful solution.

### 3.3.2 INPUT-LEVEL PERTURBATION: ORTHOGONALITY AND SYNERGY WITH DECODING STRATEGIES

SPA differs fundamentally from decoding-level randomization strategies. Techniques such as temperature sampling ($P(\text{token}|\text{context}) \propto \exp(\text{logit}/T_s)$), top-k, and top-p sampling Holtzman et al. (2020) operate on the output probability distribution derived from a single, fixed prompt. In contrast, SPA operates at the input level, generating multiple distinct prompt contexts $p_i$ *before* decoding begins for each. Each $p_i$ then conditions a unique generative process, making SPA an orthogonal approach to these decoding methods.

This orthogonality suggests a potential for synergy. For example, each SPA-generated $p_i$ could be decoded using stochastic strategies like temperature sampling ($T_s > 0$), yielding multiple outputs (e.g., $y_{i,a}, y_{i,b}, \dots$) for each distinct input. This hierarchical approach—SPA at the prompt level, stochastic methods at decoding—can enable a more thorough exploration of the solution space than either technique alone, as SPA's input modulation influences token probabilities from the start of the generative path.

With the SPA methodology established, the subsequent section introduces the `SPARC-Debugger`, a framework applying these principles to Verilog automated debugging and code generation.

## 4 THE SPARC-DEBUGGER FRAMEWORK

The `SPARC-Debugger` is the automated framework we designed to instantiate and evaluate SPA in practice. Beyond serving as a validation platform, it also contributes a structured methodology for integrating prompt perturbations with compiler-guided refinement in the Verilog domain. Specifically, it embeds SPA into a closed-loop process that couples hierarchical error-pattern matching with prompt diversification, enabling systematic exploration and correction of RTL errors.

This framework thus plays a dual role: (1) it provides a concrete instantiation of SPA, showing how lightweight input perturbations can be operationalized in realistic HDL workflows, and (2) it offers an experimental testbed for rigorously measuring SPA's impact under controlled feedback conditions. The following subsections detail its components, beginning with compilation verification and hierarchical error matching.

### 4.1 COMPILATION VERIFICATION: COMBINING HIERARCHICAL MATCHING WITH SPA

This initial stage of the `SPARC-Debugger` focuses on resolving syntax and structural errors in Register Transfer Level (RTL) code, such as missing semicolons or mismatched `begin`/`end` blocks. This stage, depicted in Figure 2(a), automates error correction by employing SPA to improve LLM-based code refinement.

The process begins with **Error Detection and Parsing**: an HDL compiler flags an error in the Hardware Design (HD), generating Error Logs. A Rule-Based Error Message Splitter then segments these diagnostic messages for detailed inspection. Subsequently, **Hierarchical Library-Based Error**

**Matching** is performed. Parsed error logs, containing an error signature $E_{new} = (e_c, e_m, e_l, e_{ids})$ (error code, message, line, identifiers), are matched against a pre-constructed **Pattern Library** ($\mathcal{L}_P$) for recurring structural faults and an **Error Library** ($\mathcal{L}_E$) for specific, known syntax issues. Crucially, both $\mathcal{L}_P$ and $\mathcal{L}_E$ are designed to be dynamic; they become increasingly comprehensive as more diverse errors are processed. This continuous learning enhances their utility over time. The matching algorithm is hierarchical:

1. **Tier 1 (Exact Error Code Matching):** The error code $e_c$ from $E_{new}$ is directly matched against error codes $c_k$ within entries $L_{E,k} = (c_k, t_k, s_k)$ in the Error Library $\mathcal{L}_E$. A match occurs if $e_c = c_k$.
2. **Tier 2 (Regex Pattern Matching):** If Tier 1 yields no match or $e_c$ is generic, the error message $e_m$ is compared against predefined regular expression patterns $R_j$ stored in the Pattern Library $\mathcal{L}_P$. A match is found if $\text{RegexMatch}(e_m, R_j) = \text{True}$.
3. **Tier 3 (TF-IDF Cosine Similarity Matching):** For errors unresolved by prior tiers, the new error message $e_m$ is compared with textual descriptions $t_k$ in libraries ($\mathcal{L}_E, \mathcal{L}_P$) Manning et al. (2008). Both $e_m$ and $t_k$ are converted into TF-IDF vectors, $\vec{V}(e_m)$ and $\vec{V}(t_k)$, respectively. The Cosine Similarity is then computed as $\text{Sim}_{\cos}(e_m, t_k) = \frac{\vec{V}(e_m) \cdot \vec{V}(t_k)}{||\vec{V}(e_m)|| \cdot ||\vec{V}(t_k)||}$. A fuzzy match is identified if this similarity score exceeds a predefined threshold $\theta_{\text{sim\_cos}}$.

If a match is identified, an Information Extractor isolates relevant details. These details, combined with prompt templates, are used to construct a targeted base prompt $p_0$. If no library-guided match is found, a general base prompt $p_0$ is constructed directly from the error log. This base prompt $p_0$ then proceeds to the **LLM Refinement** stage. Here, the **SPARC module** implements the SPA methodology by generating $N$ prompt variants $p_i$ from $p_0$ using selected noise injection strategies (PRN, SRN, KIN). Each variant $p_i$ is sent to an LLM agent, yielding $N$ candidate code corrections $y_i$. These candidates then undergo **Selection, Iteration, and Loop Control**. All $y_i$ are recompiled. If one or more compile successfully, a selection strategy determines the refined HD. If all fail, the shortest error log not indicating compiler abandonment is chosen to construct a new $p_0$ for another refinement iteration. This loop is constrained by a maximum iteration count ($Max_{compile}$).

## 4.2 FUNCTIONAL VALIDATION: SPA-DRIVEN HYPOTHESIS AND SOLUTION GENERATION

Once the HD is syntactically correct, this phase evaluates its functional correctness using a Testbench (TB), as illustrated in Figure 2(b). This phase employs SPA when LLM intervention is needed to resolve functional issues. The process initiates with a **TB-HD Interface Check**. After interface resolution, the Simulation Tool applies test vectors to the HD. If the TB simulation fails, the discrepancies trigger an error analysis. A base prompt $p_0$ is formulated using the collected failure information. The **SPARC module** then generates $N_a$ prompt variants from $p_0$. These variants guide LLM agents in inferring potential error causes, ultimately yielding a set of root cause hypotheses, $H = \{h_1, \ldots, h_{N_a}\}$.

Subsequently, for each hypothesis $h_k \in H$, a targeted base prompt $p'_{0,k}$ is created. The **SPARC module** generates $N_s$ variants $p'_{k,l}$ from $p'_{0,k}$. LLM agents then use these variants to produce $N_s$ candidate HDL modifications (fixes), $y_{k,l}$. These candidates first undergo screening for compilability. Successfully compiled candidates are re-simulated with the full TB. If a candidate passes all test cases, it is selected as the validated solution. If only partial improvements are achieved, the outcomes inform the base prompt for a subsequent iteration. This functional debugging loop is constrained by $Max_{compile}$. A Hardware Design that successfully passes all simulation vectors is considered validated.

## 4.3 EXTENDING THE FRAMEWORK FOR VERILOG CODE GENERATION

To evaluate the versatility of the SPA methodology, we extend the application of the SPARC-Debugger beyond debugging to Verilog code generation. From a high-level design specification, the SPARC module generates diverse prompt variants to elicit varied initial Verilog drafts from LLMs. These drafts then undergo iterative refinement using the SPARC-Debugger's compilation and runtime validation loops, automatically correcting syntactical, structural, and functional errors to transform initial outputs into validated hardware designs. The empirical performance of this SPA-enhanced code generation approach is detailed in the evaluation section.

## 5 EVALUATION

To validate the SPA Hypothesis and assess the efficacy of our Stochastic Prompt Assistance (SPA) methodology, this section details the experimental setup, evaluation framework, and results for Verilog debugging and code generation tasks.

### 5.1 EXPERIMENTAL SETUP

**Models and Deployment.** Our experiments utilized a diverse set of Large Language Models (LLMs), encompassing both closed-source models like OpenAI's GPT-4o and GPT-4o-mini OpenAI (2024), as well as open-source models such as Meta's Llama 3.3 Meta AI (2024) and Alibaba Cloud's Qwen3 Yang et al. (2025). To ensure fair comparisons, all models were run with consistent decoding parameters (e.g., temperature 0.6). The experiments were conducted on a server equipped with eight NVIDIA H200 GPUs. We utilized vLLM Kwon et al. (2023) and Ollama Ollama Team (2025) to deploy the local open-source models.

**Benchmarks.** We evaluate on two complementary benchmarks for Verilog tasks. For debugging, we adopt the CirFix dataset Ahmad et al. (2022), which consists of 32 defect scenarios across 11 Verilog projects. Consistent with prior work Ahmad et al. (2022); Laeufer et al. (2024), we conduct a focused evaluation on a subset of seven modules that capture the lightweight to medium-complexity designs where most common bug patterns (e.g., incorrect assignments, missing resets, sensitivity list errors) occur. This setting provides a controlled yet meaningful comparison, while leaving large IPs (e.g., i2c, sha3) to future work. Each defect is accompanied by a functional testbench for correctness checking.

For Verilog code generation, we adopt the VerilogEval benchmark Liu et al. (2023), which consists of 156 tasks collected from HDLBits. The tasks span from simple combinational circuits to complex finite state machines, and evaluate both Code Completion and Spec-to-RTL.

Together, these two benchmarks offer a complementary and standardized basis for assessing LLMs on both bug fixing and synthesis in RTL.

**Methodology and Baselines.** Our evaluation is grounded in a fair comparison under an equal computational budget. The baseline consists of an LLM using a standard, unperturbed prompt with stochastic decoding. To ensure robustness, all reported results are the average success rate over five independent runs. We focus on the single-turn, zero-shot pass@1 metric Chen et al. (2021) to directly measure the impact of each noise strategy. This setup is designed to precisely isolate SPA's contribution rather than to compete with multi-turn SOTA frameworks.

**SPA Configurations.** We evaluated the three noise types defined in our methodology: Pure Random Noise (PRN) from the full ASCII set, Structured Random Noise (SRN) using pure symbols, and Knowledge-Influenced Noise (KIN) with 20 predefined Verilog-irrelevant comment sentences. Noise was injected by prepending (Front), appending (End), or inserting into a random line as a comment (Rand). Noise lengths ($L$) were explored as percentages of the base prompt's length.

### 5.2 EVALUATION FOR VERILOG DEBUGGING

This subsection presents the empirical results of applying the SPA methodology to Verilog debugging tasks from the CirFix dataset.

*SPA Boosts Debugging Success Across Models.* The results in Figure 3 consistently demonstrate SPA's ability to improve debugging performance. For Llama 3.3, a capable but not top-tier model, the baseline success rate of approximately 63% was significantly improved by several SPA strategies, with PRN+End achieving a peak performance of around 73%—a 10% absolute improvement. The more powerful models also benefit: GPT-4o-mini shows a notable 7% absolute gain with the same PRN+End strategy, while GPT-4o, already very strong on this benchmark, still registers a smaller but positive improvement. These results suggest that SPA is broadly beneficial, though its relative impact is more pronounced for models with greater headroom for improvement. The interaction between SPA and high-capability models is more nuanced, highlighting the importance of selecting perturbation strategies thoughtfully.

*SPA Outperforms Representative Repair Frameworks.* To provide context for SPA's performance, Figure 4 compares its results against several representative repair baselines: CirFix Ahmad et al. (2022), RTL-Repair Laeufer et al. (2024), and GPT-4 VarB Ahmad et al. (2024). We stress that

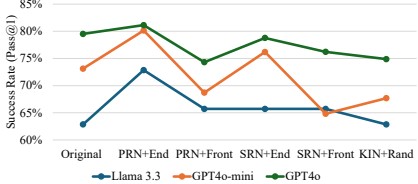

Figure 3: Overall success rate (*pass@1*) for Verilog debugging across different SPA strategies on Llama 3.3, GPT-4o-mini and GPT-4o.

Figure 4: Task-level success rate (pass@1) on the CirFix debugging benchmark, comparing SPA-enhanced LLMs with representative repair frameworks (illustrative, not strictly comparable).

this comparison is illustrative rather than a strict benchmark: CirFix and RTL-Repair are domain-specific repair systems, while GPT-4 VarB is a prompting-based study under different experimental settings, whereas our results are LLMs enhanced by SPA. Despite some variance on individual tasks, the overall row shows a consistent trend: all three SPA-enhanced models exceed the baselines (baselines ≈0.58/0.70/0.64 vs. Llama 3.3 +SPA ≈0.73, GPT-4o +SPA ≈0.82, GPT-4o-mini +SPA ≈0.82). This indicates that SPA can elevate general-purpose LLMs to be competitive with, and in some cases surpass, specialized repair tools. It highlights SPA's role as a lightweight and complementary enhancement to existing workflows rather than a monolithic replacement. All systems are evaluated under their standard settings reported by their respective papers; we do not enforce equal tool-call budgets across heterogeneous frameworks.

*SPA Breaks Persistent Failure Modes.* A more granular view for Llama 3.3 in Figure 5 shows where SPA's value is most evident. While SPA configurations remain strong on tasks where the baseline already succeeds (e.g., decoder, mux), their advantage is clearest on modules where the model gets stuck. The most striking example is the sdram task: baseline prompting yields 0% pass@1, indicating the model

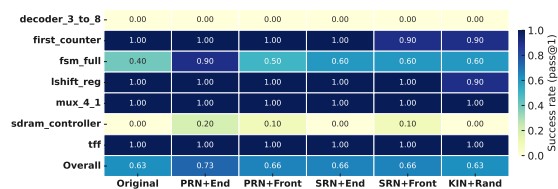

Figure 5: Task-level *pass@1* success rate for Llama 3.3 on CirFix debugging benchmark.

is trapped in a flawed generative trajectory, whereas multiple SPA strategies break this failure mode—PRN+End alone reaches 20% pass@1. This "0-to-1" unlock phenomenon is not unique to a single model; for instance, SPA also raises GPT-4o-mini's success on lshift_reg from 0% to 12% (see Appendix A for full per-model results).

These task-wise recoveries support the SPA Hypothesis that controlled syntactic perturbations can jolt a model out of brittle reasoning loops. In practical workflows, this translates into substantial ensemble gains: if an ensemble of 10 attempts used the best-performing strategy (PRN+End), the estimated pass@10 on sdram would reach approximately 89.3%. In this evaluation, we deliberately isolate a single SPA turn to measure its contribution precisely; running the full iterative SPARC-Debugger would likely obscure the specific benefit of SPA because the framework's additional components solve many cases outright. Nevertheless, these results suggest that embedding SPA-style ensembles within the full debugger can further elevate success on the most challenging debugging tasks.

## 5.3 EVALUATION FOR VERILOG CODE GENERATION

This subsection evaluates SPA's efficacy for Verilog code generation on the VerilogEval benchmark.

*Optimal SPA Strategy is Task- and Model-Specific.* The impact of SPA on code generation, as detailed in Table 1, reinforces that the optimal perturbation strategy is highly dependent on the specific model and task. For instance, PRN+Front works best for Qwen3-32B, while GPT-4o-mini benefits most from SRN+End. This contrast with debugging highlights SPA's different role: in debugging, SPA breaks persistent failure modes; in code generation, where baseline performance is already strong, its role is to expand the solution space with a portfolio of diverse candidates. This underscores the value of the full SPARC-Debugger, which can leverage ensembles of high-performing SPA variants instead of a single, fixed strategy. Since many individual strategies already

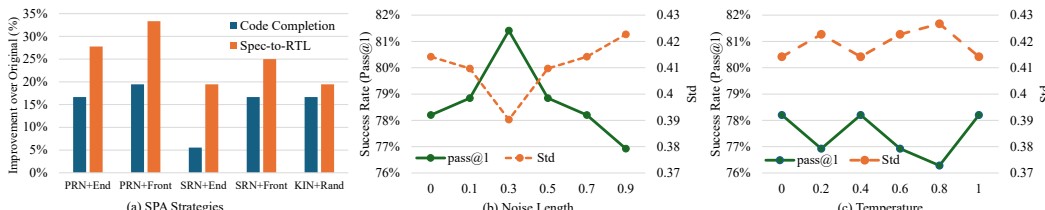

Figure 6: (a) How often the different SPA strategies improve accuracy over Original prompt on Qwen3:32B. (b) Ablation on noise length $L$ for SRN+End on GPT-4o-mini. (c) Ablation on temperature for the base prompt on the same task.

Table 1: Pass@1 success rate for VerilogEval code generation tasks under different prompt-noise strategies with different LLMs. Bold numbers mark the best score of the row.

| Task | Model | Original | PRN + End | PRN + Front | SRN + End | SRN + Front | KIN + Rand |
|------|-------|----------|-----------|-------------|-----------|-------------|------------|
| Code-Comp | GPT-4o-mini | 0.769 | 0.762 | 0.768 | **0.785** | 0.762 | 0.754 |
| Code-Comp | Llama 3.3 | **0.750** | 0.708 | 0.703 | 0.739 | 0.684 | 0.692 |
| Code-Comp | Qwen3-32B | 0.812 | 0.812 | **0.829** | 0.707 | 0.812 | 0.608 |
| Spec-to-RTL | Qwen3-32B | 0.781 | 0.769 | **0.800** | 0.744 | 0.793 | 0.657 |

yield high success rates ($> 0.75$), combining them in the debugger's iterative loop further raises the probability of finding a correct solution.

*SPA as an Orthogonal and More Effective Exploration Method.* Deeper insight into SPA's utility is provided in Figure 6. The analysis in Figure 6(a) shows that SPA strategies often succeed on Qwen3-32B tasks where the baseline fails, improving accuracy on approximately 30–33% of Code Completion cases and over 25% of Spec-to-RTL tasks. This highlights SPA's ability to generate alternative reasoning paths that the original prompt cannot access, making it an orthogonal complement to standard prompting. Further ablation on GPT-4o-mini (Figures 6(b,c)) reveals that SPA consistently outperforms conventional decoding-level randomness. Specifically, SRN+End achieves its peak performance at a moderate injection length ($L \approx 0.3$), with a robust range of $0.1 \leq L \leq 0.5$ where performance is reliably enhanced. In contrast, adjusting decoding temperature alone reaches a lower peak (about 78.3%) and exhibits unstable variance. This direct comparison confirms that SPA provides a distinct and more controllable exploration dimension, yielding more consistent gains than sampling-based diversity.

*Syntactic Noise Outperforms Semantic Noise.* Our evaluation also clarifies why syntactic perturbations succeed where semantic ones fail. As shown in Figures 3 and Table 1, Knowledge-Influenced Noise (KIN) consistently underperforms. Injecting task-irrelevant but semantically meaningful tokens distracts model attention and degrades output quality Vaswani et al. (2017). This contrast reinforces the SPA Hypothesis: lightweight syntactic noise broadens reasoning trajectories without introducing misleading semantics, whereas semantic noise destabilizes the generation process.

*Practical Lessons Learned.* Taken together, these ablations provide concrete guidance: (1) syntactic perturbations (PRN, SRN) are the most reliable choices; (2) injecting noise at 10–30% of prompt length strikes the best balance between diversity and stability; and (3) SPA serves as an orthogonal, budget-friendly complement to decoding randomness, making it broadly applicable for boosting code generation tasks under fixed inference constraints.

## 6 CONCLUSION

This work introduced and validated **Stochastic Prompt Assistance (SPA)**, a lightweight, input-level methodology that leverages an LLM's prompt sensitivity to diversify reasoning trajectories. Our empirical results across Verilog debugging and code generation benchmarks show that injecting controlled, non-semantic perturbations consistently yields pass@1 gains and, critically, achieves $0 \rightarrow 1$ unlocks where baseline models fail entirely. We further demonstrate that SPA serves as a complementary and orthogonal enhancement to decoding-time sampling and iterative refinement strategies, offering a computationally efficient means to "unstick" models from brittle trajectories. These findings validate the practical utility of the `SPARC-Debugger` and suggest that controlled input perturbation is a principled and promising method for improving the reliability of LLMs in HDLs, with potential extensions to other high-precision formal reasoning domains.

## ETHICS STATEMENT

This work does not involve human subjects, personally identifiable information, or sensitive user data. All experiments are conducted on publicly available datasets (VerilogEval and CirFix), which have been previously released for research use. Our contributions are methodological and algorithmic; we do not foresee direct negative societal impacts beyond the general risks associated with code-generation systems. We follow the ICLR Code of Ethics in ensuring integrity, fairness, and transparency throughout the research.

## REPRODUCIBILITY STATEMENT

We ensure reproducibility by using two publicly available benchmarks, VerilogEval and CirFix, described in Sections 5.2 and 5.3. Our methodology (SPA and the `SPARC-Debugger`) is detailed in Section 3, with ablations in Section 5.3 clarifying parameter choices. An anonymous GitHub repository is available at `https://anonymous.4open.science/r/Noise-Injection-to-LLM-BBED`, which contains source code and a README with usage instructions to support independent verification.

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

## A  DEBUGGING BENCHMARK RESULTS USING GPT-4O AND GPT-4O-MINI.

To evaluate the generalizability of the SPA methodology on state-of-the-art proprietary models, we performed a comprehensive evaluation using GPT-4o and GPT-4o-mini. The detailed task-level success rates (*pass@1*) are presented in Figure 7 and Figure 8. These results confirm that SPA remains a valuable technique even when applied to models with very high baseline performance.

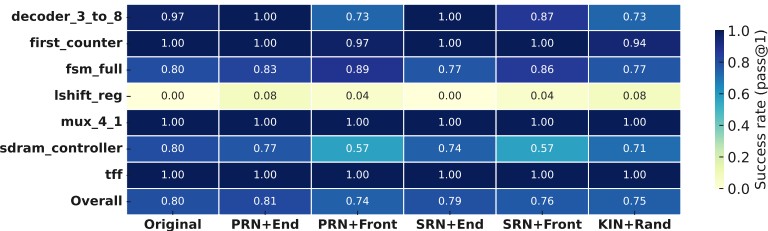

Figure 7: Task-level *pass@1* success rate for GPT 4o on CirFix debugging benchmark.

Figure 8: Task-level *pass@1* success rate for GPT 4o-mini on CirFix debugging benchmark.

Our analysis of these models yields two primary insights:

*Effectiveness on Strong Baselines.* For a high-performing model like GPT-4o, which already achieves a baseline success rate of approximately 80%, the best SPA strategy (PRN+End) further improves the overall performance to 81%. This demonstrates that even on highly capable models, SPA can provide a marginal but valuable performance gain.

*Crucial Role in Overcoming Failure Modes.* The value of SPA is most pronounced in its ability to break persistent failure modes. This is clearly demonstrated in the GPT-4o-mini results. For the challenging lshift_reg task, the baseline model completely fails, achieving a 0% success rate. However, the PRN+End strategy successfully overcomes this impasse, achieving a 12% success rate. This "0-to-1" improvement underscores SPA's core function: it is not merely a performance booster but a critical exploration mechanism that can unlock solutions inaccessible to the standard prompting approach.

We also note that certain SPA variants, particularly those introducing more semantic noise (e.g., KIN+Rand), can sometimes result in a slight performance decrease compared to the strong baseline. This is consistent with our hypothesis that SPA functions by perturbing the model's reasoning path. When a model's default path is already highly effective, some perturbations can naturally be less optimal. This reinforces the view of SPA as a portfolio of diverse strategies; its strength lies in the ensemble's ability to significantly increase the probability of finding a correct solution, especially on the most difficult tasks where the baseline model is stuck.

## B  LLM USAGE

Large Language Models (LLMs) were used as research tools during this project. Specifically, they assisted in preliminary exploration of prompting behaviors and in limited drafting support for writing. However, all technical ideas, experiments, analysis, and final writing decisions were made by the authors. No LLM is considered a contributor to the research.

