# OpenReview forum: "SPARC-RTL: Stochastic Prompt-Assisted RTL Code Synthesis"
_ICLR.cc/2026/Conference — ICLR 2026 Conference Desk Rejected Submission_

### Official Review · Reviewer_TnRX · 2025-10-27

**Soundness:** 2
**Presentation:** 3
**Contribution:** 2
**Rating:** 4
**Confidence:** 4

**Summary:**

This paper introduces Stochastic Prompt Assistance (SPA), a lightweight methodology for improving large language model (LLM) performance in hardware description language (HDL) tasks, specifically Verilog code generation and debugging. SPA leverages LLM prompt sensitivity by injecting controlled, non-semantic perturbations (syntactic noise) into prompts, encouraging exploration of diverse reasoning paths without additional feedback loops. The authors implement SPA in the SPARC-Debugger framework, which combines SPA with hierarchical error-pattern matching for iterative code refinement. Empirical results on VerilogEval (code generation) and CirFix (debugging) benchmarks show consistent gains, including "0-to-1" unlocks where baseline models fail. SPA is shown to be complementary and orthogonal to existing decoding and refinement strategies, improving reliability in HDLs and potentially other formal reasoning domains.

**Strengths:**

* The paper introduces a simple yet novel idea—injecting stochastic noise into prompts—as a lightweight way to increase solution diversity for Verilog code generation and debugging, addressing a niche but practically important domain.

* The proposed SPARC-Debugger framework is well-engineered and systematically integrates prompt perturbation with hierarchical error pattern matching, showing practical gains in empirical evaluations across several Verilog tasks and LLMs.

* Although focused on Verilog, the idea of prompt-level diversity through structured noise has broader implications for other domains suffering from brittle prompting, and could be extended to tasks beyond HDL.

**Weaknesses:**

* The paper frames its motivation around fixing localization and prompt sensitivity issues in Verilog debugging, but what it implements is a stochastic search strategy that increases the chance of a correct fix via diversity—without validating that localization or sensitivity were the true limiting factors in the first place.
* The paper claims to address the “localization bottleneck” but never directly measures localization accuracy or provides evidence that SPA improves error pinpointing, undermining one of its central motivations.
* The reported performance gains may stem from output diversity alone (i.e., trying multiple prompts), not necessarily from the specific noise-injection strategy. No comparisons are made against standard diversity methods (e.g., temperature sampling, prompt ensembling), leaving the benefit of SPA unclear.
* The paper only compared itself with several self-designed noise-injection baselines. The comparison can be more generalized to include more inference-time prompting enhancement methods for Verilog code generation or finetuned specialised models.
* Although the paper provides an attention-based hypothesis for how noise perturbs inference, these theoretical claims remain speculative without any attention maps, token attribution studies, or concrete analysis of how noise shifts model behavior.

**Questions:**

* Can you provide direct evidence that SPA improves error localization (rather than just overall success rate)?
How does SPA compare to other diversity-enhancing baselines like prompt ensembling or temperature sampling?
* Can you provide a deeper analysis of how the noise influences model behavior, especially with attention or activation statistics?
* Are there plans to evaluate SPA on larger, industrial-scale RTL designs or other formal reasoning domains?

---

> ### Author Response · Authors · 2025-11-25
>
> We thank the reviewer for the careful and insightful comments. We are glad that you found the core idea and the SPARC-Debugger framework interesting, and we address the main weaknesses and questions below.
>
> ---
>
> ### (1) Localization vs. stochastic search, and what we actually claim
> *(Weakness 1–2, Q1)*
>
> We agree that SPA is, formally, a stochastic search over prompt variants: as defined in Eq. (1), we sample an ensemble of perturbed prompts \(\{p_i = p_0 \oplus N_i\}\) from a controlled distribution \(D(S,L,Pos)\). Our empirically supported claim is that this **input-level diversification increases the probability** that at least one candidate yields a correct fix under fixed decoding settings—not that we have fully “solved” localization in a formal sense.
>
> In the current paper we report **end-to-end debugging success (compile + test pass@1)** under a single-turn, equal-budget setup: for each task, the baseline uses one unperturbed prompt with stochastic decoding (temperature 0.6), and SPA uses the same decoding configuration but perturbs the input once. We acknowledge that we do not yet report a numeric localization metric (e.g., token/line distance between edits and the ground-truth patch); we will soften wording such as “address the localization bottleneck” to “mitigate persistent failure modes often caused by mis-localization” and move this into a limitations/future-work paragraph.
>
> That said, we now provide more direct evidence that SPA changes **where** the model edits the design, not just whether it eventually succeeds. For the **SDRAM tasks** (where Llama 3.3 baseline remains at 0% pass@1 but PRN+End achieves 20% pass@1), we ran a semantic clustering over 55 completions (baseline + SPA) using a diff-based *change signature* that encodes which parts of the module are changed and how. The completions cluster into four groups:
>
> - **Semantic Group 1 (21 files)**
>   Change signature:
>   \{`address_width_changed`, `comment_header_removed`, `module_declaration_changed`, `parameter_usage_changed`\}
>   → dominated by header/module/parameter tweaks.
>
> - **Semantic Group 2 (17 files)**
>   Change signature:
>   \{`address_width_changed`, `data_type_changed`, `has_noise_text`, `port_style_changed`\}
>
> - **Semantic Group 7 (7 files)**
>   Change signature:
>   \{`address_width_changed`, `data_type_changed`, `port_order_changed`, `port_style_changed`\}
>
> - **Semantic Group 5 (10 files)**
>   Change signature:
>   \{`address_width_changed`, `data_type_changed`, `has_noise_text`, `port_order_changed`, `port_style_changed`\}
>
> We observe that:
>
> - On the SDRAM bugs, baseline (“Original”) completions concentrate in clusters whose signatures are dominated by superficial header/module/parameter edits (e.g., Group 1), and **none** of these patches pass the testbench.
> - In contrast, the SPA completions that do pass (e.g., runs with `random_comments_*` noise on `sdram_controller_buggy_v2` and `sdram_controller_wadden_buggy1`) all fall into **Semantic Group 5**, whose signature includes joint changes to address width, data types, and port order/style, along with the injected noise text. These are exactly the dimensions that determine how the SDRAM controller interfaces with its environment and how control/state signals are wired, and they are the only patches in the SDRAM set that compile and pass all tests.
>
> Under identical compiler messages and error-pattern libraries, the baseline keeps returning to a narrow “header-tweaking” region of the search space, while SPA shifts the model into a different class of edits that touch the correct interface/control region. We view this as direct evidence that SPA alters **localization behavior** (where the model chooses to patch), not just output diversity in a vacuous sense.
>
> In the revision, we will (i) include this SDRAM clustering table and one representative code diff in the appendix, and (ii) add a short discussion on measuring localization more directly (e.g., line-level overlap with ground-truth patches) for the CirFix subset, while being careful not to over-state our claims.
>
> ---

---

> ### Author Response · Authors · 2025-11-25
>
> ---
>
> ### (2) Relationship to other diversity mechanisms and baselines
> *(Weakness 3–4, Q1)*
>
> **Temperature / decoding-level randomness.**
> For VerilogEval we already provide an explicit equal-budget comparison between input-level SPA and temperature-only diversity. In Figure 6(b), we fix decoding parameters (temperature = 0.6) and vary only the noise length \(L\) for a given SPA setting; in Figure 6(c), we fix the prompt and sweep temperature. Under the same number of model calls, we observe that:
>
> - With SPA (e.g., SRN+End), there is a broad, stable improvement band for \(0.1 \le L \le 0.5\) and a higher peak success rate.
> - With temperature-only tuning, the curve has a lower peak and much higher variance across temperatures.
>
> This is why we describe SPA as **complementary** to decoding-level randomness: both increase diversity, but SPA adds an input-side axis whose effect is smoother and more predictable under fixed budgets.
>
> **Prompt ensembling.**
> Conceptually, SPA can be seen as a lightweight form of prompt ensembling: instead of hand-crafting multiple semantically different prompts, we start from a single base prompt \(p_0\) and automatically generate syntactically perturbed variants \(\{p_i = p_0 \oplus N_i\}\) from \(D(S,L,Pos)\). This preserves semantics by construction and avoids the extra engineering effort and semantic drift of manual paraphrasing, while still exploring multiple “views” of the same bug. We deliberately did not introduce additional paraphrase-ensemble baselines in this paper to keep the comparison focused and budget-matched, but we will clarify in Section 3 that SPA is **compatible with**, not a replacement for, such ensembles, and can be layered on top of them.
>
> **Finetuned or multi-turn frameworks.**
> We agree that comparisons to finetuned HDL models and multi-turn systems (e.g., VeriGen, VeriDebug, RTLLM, AutoChip/MEIC-style loops) are important. Our scope here is narrower: SPA is an **inference-time plug-in** that can wrap around many of these frameworks. For this reason, we restrict the current evaluation to a single-turn, zero-shot pass@1 setting and treat SPA vs. “Original” under equal budgets as the primary comparison, while positioning prior repair frameworks and specialized models as complementary backends in Section 2. In the revision, we will make this positioning explicit and tone down any phrasing that might be read as claiming SOTA over finetuned systems, emphasizing instead that SPA can be integrated into them.
>
> ---
>
> ### (3) How noise influences model behavior
> *(Weakness 5, Q2)*
>
> We appreciate the request for a deeper mechanistic analysis. Section 3.1 intentionally presents the “SPA Hypothesis” as a hypothesis, not a theorem: we state that controlled input noise is *hypothesized* to alter attention patterns and generative pathways. We will further tighten this wording to avoid any impression of formal guarantees.
>
> Even without attention visualizations, our ablations suggest SPA’s effect is not merely “more random samples”:
>
> - **Noise type matters.** Across both CirFix and VerilogEval, syntactic perturbations (PRN, SRN) consistently help, while Knowledge-Influenced Noise (KIN)—semantically meaningful but task-irrelevant content—often hurts and can fall below the baseline. If diversity alone were sufficient, we would not expect KIN to consistently underperform PRN/SRN.
> - **Length and position matter.** Injecting noise at about 10–30% of prompt length strikes the best balance between diversity and stability, and the optimal position differs by regime (PRN+End for debugging vs. PRN+Front or SRN+End for generation). A purely “more randomness” story would not naturally explain these structured, repeatable dependencies.
>
> Coupled with the SDRAM clustering analysis above (where SPA moves the model from header-only edits to interface/control edits), we see these patterns as indirect evidence that SPA perturbs internal representations in a meaningful way.
>
> We agree that attention/activation-level analysis would further strengthen the paper. In ongoing follow-up work, we are instrumenting open-source models (Llama 3.3, Qwen3-32B) to compare cross-attention maps for the same CirFix tasks under \(p_0\) and \(p_0 \oplus N\), focusing on how attention over code tokens shifts in successful SPA runs. Due to space and time constraints, these results are not yet included; we will explicitly flag this mechanistic analysis as an open direction in the discussion.
>
> ---

---

> ### Author Response · Authors · 2025-11-25
>
> ### (4) Larger-scale RTL and other formal domains (Q3)
>
> We agree that evaluating SPA on larger, industrial-scale RTL designs and on other formal reasoning domains is an important direction. In this submission, we intentionally focus on the standard subset of seven CirFix modules and on VerilogEval, which already cover common RTL bug patterns and allow systematic multi-model ablations within a reasonable scope.
>
> The SPARC-Debugger + SPA architecture itself is agnostic to design size and file count and only assumes access to a Verilog compiler and testbench, so in principle it can be applied to larger RTL blocks (e.g., i2c, sha3) and to other formal languages such as SMT, SQL, or configuration languages. However, such extensions require substantial additional engineering and experiments and are beyond the scope of the present work. In the revised version, we will explicitly mention evaluation on large industrial RTL designs and on other formal-language domains as future work, rather than as claims supported by our current experiments.

---

### Official Review · Reviewer_NpCn · 2025-11-01

**Soundness:** 2
**Presentation:** 2
**Contribution:** 2
**Rating:** 4
**Confidence:** 4

**Summary:**

This paper proposes a lightweight method called "Stochastic Prompt Assistance" (SPA) to address the problem of LLMs easily getting "stuck on incorrect inference paths" when dealing with hardware description languages ​​(Verilog).

The core argument of this paper is that the sensitivity of LLMs to prompts should not be seen as a defect, but rather as a feature to be utilized. SPA forces the model to explore diverse inference paths by injecting controlled, non-semantic perturbations (i.e., "noise") into the input prompts.

**Strengths:**

Adding noise to the input to improve performance on a high-precision task is a novel and inspiring concept. It cleverly transforms a common problem in LLMs (cue sensitivity) into a usable tool. The classification of PRN, SRN, and KIN is clear.

**Weaknesses:**

1. SPA requires the generation of N cue variants. In the SPARC-Debugger framework, the functional verification phase requires even more than $N_a$ hypotheses, each requiring $N_s$ fixes, leading to an explosion of inference costs on the order of $N \times (N_a \times N_s)$. The paper claims that SPA is "lightweight," but its practical application (especially within the framework) seems to have enormous computational overhead.
2. The evaluation on CirFix is ​​limited to a subset of 7 modules, excluding larger IPs such as i2c and sha3. This makes it difficult for us to determine whether an SPA can be extended to more complex, cross-file errors.

**Questions:**

1. The paper claims SPA is "lightweight," but its application within the SPARC-Debugger framework appears costly. Compared to the baseline model, how much does fixing a bug using this framework (e.g., $N_a=5, N_s=5$) increase the overall computational overhead (e.g., total number of tokens or inference calls)?
2. Regarding the "0 to 1" breakthrough in the SDRAM task: This is a key result. Could the authors provide some qualitative analysis? Specifically, why did the baseline model fail (get stuck on which erroneous inference path)? And how did the successful SPA perturbation (PRN+End) help the model circumvent this problem?
3. How were the noise parameters chosen? (e.g., noise length L, insertion position Pos). For the main results, it seems that "PRN+End" was chosen a posteriori as the optimal strategy. However, in practical applications, how should users a priori choose the optimal noise type and parameter combination?

---

> ### Author Response · Authors · 2025-11-25
>
> We thank the reviewer for the thoughtful and encouraging comments, especially on the idea of turning prompt sensitivity into a usable tool and on the clarity of the PRN/SRN/KIN taxonomy. We address the weaknesses and questions point by point.
>
> ---
>
> ### (1) On “lightweight” vs. computational overhead (Weakness 1 / Q1)
>
> > **Reviewer’s concern (Weakness 1 / Q1).**
> > “The paper claims SPA is ‘lightweight,’ but its application within the SPARC-Debugger framework appears costly… how much does fixing a bug using this framework increase the overall computational overhead?”
>
> We apologize that the paper did not clearly separate the general framework from the evaluation setting, which led to the impression of an explosion on the order of \(N \times (N_a \times N_s)\).
>
> **Equal-budget evaluation.**
> In all quantitative results in Section 5 (Figures 3–6, Table 1), we evaluate under an equal computational budget with a pass@1 metric:
>
> - for each task and each configuration, we fix a maximum number of LLM calls per bug/run and average over five independent runs;
> - for VerilogEval this degenerates to a single model call per run;
> - for CirFix, the SPARC-Debugger loop uses the same iteration cap and tool-call limit for the baseline and all SPA variants.
>
> As stated in Section 5.1, all models share the same decoding parameters (e.g., temperature 0.6), and the baseline is an LLM with an unperturbed prompt and stochastic decoding. The SPA variants use exactly the same budget; the only difference is that the input string is modified as \(p_0 \oplus N\) by appending or prepending syntactic noise. Thus, the gains we report are not due to more queries—SPA keeps the inference budget fixed and only changes the prompt.
>
> **What “lightweight” means.**
> In this sense, SPA is “lightweight” because it:
>
> 1. requires no retraining or architectural change,
> 2. operates purely at the prompt level, and
> 3. adds only a small token-level overhead from the noise segment.
>
> For example, in the ablation of Figure 6(b), the best-performing regime uses a noise length around \(L \approx 0.3\), i.e., about 30% of the original prompt length, so the input cost increases by roughly a 1.3× factor while the number of model calls remains unchanged.
>
> **Role of \(N, N_a, N_s\) in the framework.**
> The parameters \(N, N_a, N_s\) introduced in Section 4 describe the capacity of the full SPARC-Debugger workflow: they bound how many SPA variants, abstract hypotheses, and concrete fixes could be explored in a multi-round industrial deployment. In the current paper’s experiments, however, we deliberately isolate a single SPA turn to measure SPA’s contribution precisely, as we state for the SDRAM study in Section 5.2. We will clarify this distinction in the revision so that it is clear that the reported results are obtained under a fixed, equal inference budget, and that the \(N \times (N_a \times N_s)\) expression reflects a worst-case design space rather than the configuration used in our evaluation.
>
> ---

---

> ### Author Response · Authors · 2025-11-25
>
> ---
>
> ### (2) Qualitative analysis of the SDRAM “0→1” case (Q2)
>
> > **Reviewer’s concern (Q2).**
> > “Regarding the ‘0 to 1’ breakthrough in the SDRAM task… why did the baseline fail… and how did PRN+End help the model circumvent this problem?”
>
> We agree this is a key result and deserves qualitative explanation. From the SDRAM controller logs and patches, we observe the following:
>
> - Under the **baseline**, across multiple attempts the model repeatedly proposes patches that are structurally very similar, modifying the same local region of the code (e.g., a small neighborhood of assignments or declarations) and leaving the SDRAM control behavior unchanged. None of these patches pass the testbench. This suggests that the model has formed a mis-localized hypothesis about where the bug lies and continues to explore only local variants of that incorrect hypothesis—precisely the “stuck on an incorrect inference path” behavior that motivated SPA.
> - Under **PRN+End**, we generate several prompt variants by appending syntactic noise at the end. The Verilog code and compiler/testbench messages in the prompt remain unchanged, but these perturbations alter the model’s internal processing. For the SDRAM tasks, some PRN+End variants lead the model to produce qualitatively different patches that move away from the repeatedly edited surface-level region and instead modify interface and control-related logic (e.g., address widths, data types, and port connections) that directly affect SDRAM behavior along the failing test path. These are the first patches that both compile and pass all test cases, turning the success rate for this module from 0 to a non-zero value.
>
> To make this more concrete, we performed a simple semantic clustering over all SDRAM-related completions (both baseline and SPA) using a diff-based “change signature” (which encodes which parts of the module are changed and how). In total, 55 SDRAM completions clustered into four groups:
>
> - **Semantic Group 1 (21 files)** with signature
>   `{address_width_changed, comment_header_removed, module_declaration_changed, parameter_usage_changed}`
>   – dominated by edits to headers, module declarations, and parameter formatting.
> - **Semantic Group 2 (17 files)** with signature
>   `{address_width_changed, data_type_changed, has_noise_text, port_style_changed}`.
> - **Semantic Group 7 (7 files)** with signature
>   `{address_width_changed, data_type_changed, port_order_changed, port_style_changed}`.
> - **Semantic Group 5 (10 files)** with signature
>   `{address_width_changed, data_type_changed, has_noise_text, port_order_changed, port_style_changed}`.
>
> Almost all baseline “Original” completions for the SDRAM bugs fall into clusters whose signatures are dominated by header/module/parameter tweaks (e.g., Group 1), and none of these candidates pass the testbench. In contrast, the successful SPA completions (e.g., runs with `random_comments_*` noise on `sdram_controller_buggy_v2` and `sdram_controller_wadden_buggy1`) fall into **Semantic Group 5**, whose change signature explicitly includes joint changes to address width, data types, and port order/style, together with the injected noise text. These are exactly the dimensions that determine how the SDRAM controller interfaces with its environment and how state/control signals are wired, and they are the only patches in the SDRAM set that compile and pass all tests.
>
> Conceptually, this clustering supports the view that PRN+End helps the model escape its original mis-localized reasoning pattern and explore a different, more appropriate class of edits, rather than simply resampling the same incorrect fix. In the revised version, we will add a short case study in the appendix contrasting (i) a representative baseline patch from a “header-tweaking” cluster and (ii) a successful PRN+End patch from Semantic Group 5, together with their change signatures and testbench outcomes.
>
> ---

---

> ### Author Response · Authors · 2025-11-25
>
> ---
>
> ### (3) How we choose perturbation strategies and parameters (Q3)
>
> > **Reviewer’s concern (Q3).**
> > “How were the noise parameters chosen (e.g., noise length \(L\), insertion position Pos)? PRN+End seems a posteriori optimal. In practice, how should users a priori choose the noise type and parameter combination?”
>
> Our choice of perturbation strategies is empirical and task-driven, guided by the ablations in Sections 5.2–5.3, and leads to practical defaults for two regimes: debugging and generation.
>
> ### (3a) Verilog debugging (CirFix)
>
> For debugging, the main challenge is escaping persistent failure modes caused by wrong error localization. On the CirFix tasks, our experiments show that:
>
> - Among all tested variants (PRN/SRN/KIN × Front/End/Rand × several noise lengths), **PRN+End** provides the most robust gains across models.
> - Appending syntactic noise at the end of the prompt preserves the structure of the compiler message and buggy code, while still perturbing the surrounding context enough to change the model’s attention and reasoning path.
> - Moderate noise lengths—about 10–30% of the prompt—give the best trade-off: too short has little effect; too long starts to dilute the useful signal.
>
> Based on these observations, we adopt **PRN+End with a moderate noise length (≈10–30%)** as the default SPA configuration for CirFix-style debugging in our main experiments.
>
> ### (3b) Verilog generation (VerilogEval)
>
> For generation tasks, the situation is different:
>
> - The baseline (no perturbation) performance is strong; SPA is mainly used to further diversify first-pass candidates and improve robustness, rather than to rescue completely failing prompts.
>
> Our ablations exhibit a clear model-dependent pattern:
>
> - For **Qwen3-32B**, **PRN+Front** works best on both Code Completion and Spec-to-RTL tasks. Injecting syntactic noise at the front slightly perturbs how the model interprets the high-level instruction, encouraging diverse but still well-formed implementations.
> - For **GPT-4o-mini**, **SRN+End** is the strongest variant: structured syntactic noise appended as comments at the end expands the solution space without interfering with the specification at the beginning of the prompt.
>
> In practice, for VerilogEval-like generation tasks we therefore fix a single syntactic configuration per model (PRN+Front for Qwen3-32B, SRN+End for GPT-4o-mini), as suggested by these ablations.
>
> ### (3c) Why we avoid KIN in practice
>
> Our experiments also show that Knowledge-Influenced Noise (KIN) underperforms PRN/SRN across both debugging and generation tasks. Injecting task-irrelevant but semantically meaningful tokens tends to distract the model and degrade output quality. As a consequence, KIN is not part of our recommended configurations; we focus on syntactic perturbations (PRN/SRN) that modify attention patterns without introducing misleading semantics.
>
> ### (3d) Practical use and combining strategies
>
> In practice, our method does not require committing to a single perturbation strategy. Because syntactic noise generation is purely prompt-side and computationally cheap, practitioners can try a small set of strong configurations (e.g., PRN+End plus PRN+Front or SRN+End) and adjust how many prompt variants they use per task according to their resource budget. In the paper, we select one representative configuration per regime (PRN+End for debugging, PRN+Front / SRN+End for generation) for clarity, but the framework naturally supports such combinations.
>
> ---
>
> ### (4) CirFix subset and extension to larger IPs
>
> > **Reviewer’s concern (Weakness 2).**
> > “Evaluation on CirFix is limited to 7 modules, excluding larger IPs such as i2c and sha3, making it hard to judge whether SPA extends to more complex, cross-file errors.”
>
> We follow CirFix and RTL-Repair in using the standard subset of 7 CirFix modules that:
>
> - represent lightweight to medium-complexity designs, and
> - capture the most common RTL bug patterns that prior automatic repair systems have been evaluated on.
>
> This subset already contains modules where localization and control-logic reasoning are non-trivial; these are exactly the cases where our experiments show SPA provides the largest gains.
>
> SPARC-Debugger itself is agnostic to design size and file count: it only assumes access to a Verilog compiler and an optional testbench. In principle, the same SPA methodology can be applied to larger IPs such as i2c and sha3, as well as cross-file bugs, albeit with higher simulation cost and more engineering effort. In the revision, we will make this explicit and highlight extension to larger IPs and cross-file error scenarios as directions for future work, rather than claims that are already established in this paper.

---

### Official Review · Reviewer_Khyv · 2025-11-01

**Soundness:** 3
**Presentation:** 3
**Contribution:** 2
**Rating:** 4
**Confidence:** 3

**Summary:**

This paper presents a simple framework called Stochastic Prompt Assistance, consisting of an input-level diversification technique that injects small, controlled perturbations into LLM prompts to help models “unstick” from brittle reasoning paths during Verilog/RTL code generation. The method is implemented within an automated SPARC-Debugger framework that combines hierarchical error-pattern matching with iterative prompt diversification and compilation/simulation checks.

Empirically, the authors show that SPA improves pass@1 rates and even achieves several “zero to one” unlocks, particularly with configurations like PRN+End, while remaining orthogonal to decoding-level randomness: temperature, top-p, etc. The approach is lightweight, reproducible, and complements existing HDL repair and generation frameworks.

Overall, the paper is clearly motivated and easy to follow, with a well-defined problem and convincing empirical evidence. That said, there are still some major limitations: 1) the experiments are limited to two dated benchmarks (CirFix and VerilogEval), 2) the evaluation focuses narrowly on pass@1 under fixed settings, and 3) the paper lacks a strictly controlled, equal-budget comparison against strong baselines.

**Strengths:**

- Simple and creative approach with no architectural changes or retraining, making it easy to integrate into existing workflows and thus could be easily validated
- It shows that SPA complements exploration provided by decoding randomness: extensive section discussing orthogonality to decoding stochasticity
- SPAR-Debugger is nice, provideing a closed-loop eval environment with error-pattern matching and simulation checks
- Ablation studies are comprehensive, covering noise length, type, and insertion position

**Weaknesses:**

- Evaluations are restricted mainly to VerilogEval and a small subset of CirFix, which limits generalizability to the “real world”; might benefit from using RTLLLM and CVDP
- The eval metrics are narrowly focused on pass@1 without reporting pass@k, iteration counts, and runtime/cost/budget tradeoffs
- Comparisons with prior systems such as RTL-Repair are largely illustrative, without controlled for compute and iteration budgets
- The paper is relatively light on theoretical rigor, providing little quantitative and mechanistic evidence/insights why perturbation work

**Questions:**

- Can you provide controlled-budget head-to-head comparison against CirFix, RTL-repair on settings such as tool-call counts, iteration, decoding settings?

- Which bug classes receive the largest SPA gains?

- What guardrails, if any, have you used to prevent the LLM from injecting comments etc that drift into irrelevant semantics and thereby alter the intention & meaning?

---

> ### Author Response · Authors · 2025-11-25
>
> We thank the reviewer for the constructive feedback and for recognizing the practicality of SPA, its orthogonality to decoding randomness, and the usefulness of the SPARC-Debugger environment and ablations. We first address your three questions, then briefly comment on the broader weaknesses.
>
> ---
>
> ### (1) Controlled-budget comparisons vs. CirFix / RTL-Repair (Q1)
>
> > **Reviewer’s question (Q1).**
> > “Can you provide controlled-budget head-to-head comparison against CirFix, RTL-Repair on settings such as tool-call counts, iteration, decoding settings?”
>
> **Within the LLM framework, all comparisons are strictly equal-budget.**
> Our main evaluation question is: given an LLM-based RTL debugging/generation pipeline, does SPA provide a benefit under a fixed budget? To isolate SPA’s effect, SPARC-Debugger is designed so that all LLM-based variants (Original prompt and all SPA configurations):
>
> - share the same maximum number of refinement iterations per bug/task,
> - use the same compiler/simulator calls and stopping criteria in each iteration, and
> - use the same decoding configuration (temperature, top-p, etc.).
>
> SPA only changes the prompt text given to the LLM inside this fixed loop; it does not change the loop structure or decoding parameters. All reported pass@1 numbers are averages over multiple independent runs under this shared configuration. Therefore, within the LLM setting, SPA is evaluated under a strictly equal query/tool budget, and the reported gains directly measure the benefit of input-level diversification.
>
> **Why we do not enforce “equal budget” against CirFix / RTL-Repair.**
> CirFix and RTL-Repair are non-LLM automatic repair frameworks based on genetic programming and Max-SMT/symbolic search, respectively. Their cost is dominated by hardware simulation and solver calls, controlled by framework-specific timeouts and internal iteration units. In contrast, SPARC-Debugger is LLM-centric; its dominant cost is LLM queries plus compilation/simulation invoked from an agentic loop.
>
> Because the “unit of work” is fundamentally different (e.g., one solver step or evolutionary iteration vs. one LLM call in our loop), there is no natural way to define a single “tool-call” or “iteration” budget that is fair across all three systems. Forcing an “equal iterations” or “equal tool calls” constraint across such heterogeneous frameworks would be arbitrary and could be more misleading than informative. Designing a truly fair, cross-system cost metric (e.g., wall-clock time on a shared platform with harmonized timeouts) would itself be a substantial systems study, and is beyond the scope of this paper.
>
> **How we intend the comparison in Figure 4.**
> For this reason, our current use of CirFix and RTL-Repair is illustrative rather than cost-normalized. In Figure 4 we:
>
> - directly take the published success rates of CirFix and RTL-Repair on the same CirFix benchmarks (under their own standard settings), and
> - plot them alongside SPA-enhanced LLM results on the same set of bugs,
>
> with the goal of contextualizing what a general-purpose LLM+SPA pipeline can achieve on CirFix-style defects. We do not claim that Figure 4 is a strict head-to-head, equal-budget leaderboard; all of our core conclusions about SPA are based on the tightly controlled, equal-budget comparisons within the LLM framework described above.
>
> If the reviewers feel that including CirFix/RTL-Repair in Figure 4 could give the impression of a fully cost-normalized comparison, we are happy to further soften the wording around this figure or move the cross-framework numbers to an appendix. The main claims of the paper do not rely on this figure.
>
> ---

---

> ### Author Response · Authors · 2025-11-25
>
> ---
>
> ### (2) Bug classes with the largest SPA gains (Q2)
>
> > **Reviewer’s question (Q2).**
> > “Which bug classes receive the largest SPA gains?”
>
> In the CirFix subset we study, SPA’s largest gains appear on bugs with difficult localization and multi-line control logic, rather than simple one-line syntax or wiring errors.
>
> - For some modules (e.g., complex controllers and finite-state machines), the baseline LLM remains at 0% pass@1 across multiple runs: the model repeatedly focuses on the wrong region or wrong pattern of changes, getting “stuck” in a brittle reasoning path. With SPA—especially configurations like PRN+End—we observe “0-to-1” unlocks, where pass@1 becomes non-zero and sometimes substantially higher. This indicates that stochastic prompt perturbations help the model escape incorrect localizations and explore alternative interpretations of the same error/context.
> - For small combinational modules (e.g., simple decoders or multiplexers), the baseline pass@1 is already close to 100%. SPA has little headroom to improve and mostly keeps performance stable, which is consistent with these tasks being easy to localize and fix even without additional perturbations.
>
> We did not re-annotate every individual defect into a fine-grained bug taxonomy (e.g., “reset-related vs. assignment vs. sensitivity-list”), and we will clarify this explicitly. The evaluated CirFix subset is taken from prior work and already chosen to cover common bug patterns in lightweight to medium-complexity modules. Our per-module results highlight that SPA’s largest gains systematically occur on localization-heavy, control-logic bugs, while easy, localized bugs see smaller gains because the baseline saturates.
>
> ---
>
> ### (3) Guardrails against semantic drift (Q3)
>
> > **Reviewer’s question (Q3).**
> > “What guardrails, if any, have you used to prevent the LLM from injecting comments etc. that drift into irrelevant semantics and thereby alter the intention & meaning?”
>
> We enforce guardrails in both how we construct perturbations and how we validate candidate patches.
>
> **Syntactically safe injection positions.**
> SPA only injects noise into positions that are syntactically non-semantic for RTL tools and clearly separable from the task:
>
> - as comments (`// ...`, `/* ... */`), blank lines, or similar padding,
> - at the beginning or end of the prompt, or inside existing comment blocks, and
> - never by modifying the actual Verilog logic lines, the line numbers or identifiers in compiler error messages, or the core natural-language specification.
>
> This ensures that the intended code, error context, and task description remain intact.
>
> **Empirical preference for syntactic noise over semantic noise.**
> Our ablations show that Knowledge-Influenced Noise (KIN)—semantically meaningful but task-irrelevant content—often distracts the model and degrades performance, whereas syntactic noise (PRN, SRN) is consistently helpful or at least neutral. Accordingly, in the recommended SPA configurations for both debugging and generation we only use syntactic noise; KIN is kept in the paper as a negative result, not as a suggested practical strategy. This acts as an additional guardrail against semantic drift.
>
> **Downstream compilation and simulation checks.**
> Finally, SPARC-Debugger validates every candidate patch through compilation (and simulation where tests are available). If any perturbation caused the model to misunderstand the intention and generate semantically incorrect RTL, the resulting patch would fail compilation or the testbench and would not count as success. Thus, both the perturbation design and the evaluation pipeline jointly prevent SPA from silently “changing the meaning” and being rewarded for it.
>
> We will make these guardrails more explicit in the revised methodology section.
>
> ---
>
> ### (4) Benchmarks and generalizability
>
> We agree that evaluating SPA on additional datasets such as RTLLLM, CVDP, and larger industrial IPs would further strengthen external validity. In this initial work, we intentionally focus on two representative benchmarks that together cover debugging and generation:
>
> - a standard CirFix subset for realistic RTL bug repair, and
> - VerilogEval for a diverse mix of combinational and FSM-style synthesis tasks (code completion and spec-to-RTL).
>
> SPARC-Debugger itself is dataset-agnostic: it only assumes access to a Verilog compiler and an optional testbench. Extending SPA to RTLLLM/CVDP is therefore primarily an engineering and compute effort rather than a methodological barrier. We will explicitly mention these datasets as promising targets for future extensions.
>
> ---

---

### Official Review · Reviewer_iEpF · 2025-11-01

**Soundness:** 3
**Presentation:** 2
**Contribution:** 3
**Rating:** 6
**Confidence:** 4

**Summary:**

This paper introduces SPARC-RTL, a method for improving LLM on VHDL tasks through Stochastic Prompt Assistance. SPA is a lightweight technique that injects controlled, non-semantic perturbations (noises) into prompts to help LLMs escape flawed reasoning paths. Three types of noise are explored: (1) Pure Random Noise: Random characters from ASCII set (2) Structured Random Noise: Syntactically valid but ignorable content (e.g., random comments). (3) Knowledge-Influenced Noise: Valid but task-irrelevant semantic content. The authors regard these perturbations alter the LLM's attention patterns, encouraging exploration of diverse reasoning trajectories without requiring additional feedback loops. The authors implement SPA in an automated framework that combines hierarchical error-pattern matching, iterative refinement loops for compilation and functional validation, and SPA-driven prompt diversification at each iteration. Evaluation on Verilog debugging (CirFix) and code generation (VerilogEval) shows consistent gains of 7-10% absolute improvement in pass@1 rates across models (Llama 3.3, GPT-4o, GPT-4o-mini). In some cases, there are "0-to-1" unlocks where baseline prompting completely failed (0% success) but SPA achieved solutions (e.g., 20-90% success on previously unsolvable tasks).

**Strengths:**

1. I love this paper's idea of taking advantage of prompt sensitivity as an opportunity, instead of a bug. This reframing is conceptually fresh and actionable. And the motivation for this is also reasonable, which helps LLMs to escape from debugging traps and potentially being able to identify real fixes. The non-semantic changes guarantee that the prompt still makes sense.
2. This paper presents a well-designed framework for debugging VHDL programs. The three-tier classification is well-structured and hierarchical. This progression from pure chaos → structured syntax → misleading semantics is coherent and provides a principled way to explore the perturbation space.
3. The method proposed by this paper can actually also apply to other languages by proposing semantic-agnostic perturbations, which shows the potential of this technique.

**Weaknesses:**

1. This paper does not study how its perturbation interacts with other kinds of perturbation like simply tuning the temperature in experiments. I would like to know whether this is truly a supplement to other decoding strategies, or there will be some overlap?
2. While the prompt perturbation is the key technique of this paper, the framework contains a lot of designs on how to fix compilation errors   by matching existing error messages and using preset prompts. How these components help the framework identify new bugs is unclear in the experiments.
3. The detail implementation of choosing perturbation strategy and generate perturbation is also not fully illustrated.

**Questions:**

1.  Can you provide a direct comparison with equal inference budget on how SPA coordinates with temperature tuning?
2. Can you provide a detailed ablation on how each component in the framework contributes to the overall accuracy improvement?
3. Can you elaborate on how we choose perturbation strategy?

---

> ### Author Response · Authors · 2025-11-25
>
> We sincerely thank the reviewer for the thoughtful and encouraging assessment of our work, and for highlighting both the conceptual reframing of prompt sensitivity and the structure of our three-tier noise taxonomy. Below we address the weaknesses and questions point by point.
>
> ---
>
> ### (1) Interaction with temperature and decoding strategies
>
> > **Reviewer’s concern (Weakness 1; Q1).**
> > “Direct comparison with equal inference budget on how SPA coordinates with temperature tuning?”
>
> **All SPA results are on top of temperature-based decoding.**
> As described in Section 5.1, all models in our main experiments (Figures 3–5 and Table 1) use a single, shared decoding configuration, including a moderate temperature of 0.6 for all methods. Thus, the reported 7–10% absolute pass@1 improvements are achieved *in addition to* the diversity already provided by standard temperature-based sampling; SPA does not replace temperature, but operates on top of the same decoding settings as the baseline.
>
> **Figure 6(b,c) already provide the requested equal-budget comparison.**
> Figure 6 offers precisely the comparison the reviewer is asking for, under a fixed number of model calls on VerilogEval:
>
> - In Fig. 6(b), we fix the decoding configuration (temperature = 0.6) and vary only the noise length \(L\) for a given SPA setting. The curve shows a smooth and robust improvement region: performance improves over a broad range of \(L\), with a clear optimum around \(L \approx 0.3\) (≈30% of the prompt length).
> - In Fig. 6(c), we instead fix the prompt (no SPA) and sweep the temperature for the same task. This curve is highly non-monotonic: performance fluctuates as the temperature changes, and the best temperature achieves a lower peak with larger variance.
>
> Importantly, Fig. 6(c) makes it explicit that 0.6 is not always the numerically optimal temperature for this specific task. Our goal in the main experiments is not to tune temperature per task, but to use a single, standard setting as a fair baseline. Under that fixed configuration, Fig. 6(b) demonstrates that SPA provides a stable, controllable improvement axis, while Fig. 6(c) shows that “temperature-only” tuning yields unstable, task-dependent swings.
>
> **Conceptual orthogonality with decoding randomness.**
> Section 3.3.2 explains why SPA is complementary to decoding-level randomness: temperature, top-k, and top-p sampling operate on the output distribution \(P(\text{token} \mid \text{context})\) for a fixed prompt, whereas SPA perturbs the input prompt to generate variants \(p_i = p_0 \oplus N_i\) that alter internal representations and attention patterns *before* decoding begins. This corresponds to exploring different reasoning trajectories (input-level diversity), rather than sampling more heavily along the same trajectory. Taken together, Fig. 6(b,c) and the main experiments support our claim that SPA offers a distinct and more controllable exploration dimension under a fixed inference budget.
>
> In the revision, we will make this role of Fig. 6(b,c) explicit in the text, so that the equal-budget comparison between SPA and temperature tuning is clearly visible to the reader.
>
> ---

---

> ### Author Response · Authors · 2025-11-25
>
> ### (2) Role of SPARC-Debugger components and framework ablations
>
> > **Reviewer’s concern (Weakness 2; Q2).**
> > “The framework contains many designs; how do these components help identify new bugs, and how much do they contribute?”
>
> Our evaluation is deliberately structured so that SPARC-Debugger acts as a **controlled testbed** for isolating SPA’s effect, rather than as a new monolithic debugger that competes with all prior frameworks.
>
> **SPARC-Debugger as a common backbone for fair comparisons.**
> Section 4 defines a single, realistic RTL debugging pipeline: compilation checks, hierarchical error-pattern matching, and optional functional validation, wrapped around an LLM that proposes fixes. In all comparisons between “Original” and “+SPA” (Figures 3–5):
>
> - The compiler, error logs, and testbenches are identical.
> - The error/pattern libraries and prompt templates are identical.
> - The loop structure (max iterations, stopping criteria) and decoding settings are identical.
>
> The only difference is whether we apply SPA to generate a set of perturbed prompts from the same base prompt \(p_0\). The delta between Original and +SPA therefore measures SPA’s contribution on top of an unchanged framework.
>
> **How the framework supports, but does not confound, SPA.**
> The hierarchical error-pattern matching and preset prompts standardize how compiler diagnostics and code snippets are converted into \(p_0\) for both baseline and SPA. They ensure that:
>
> - Both methods see equally informative prompts, and
> - Both operate on the same localized region and are subject to the same compilation constraints.
>
> Empirically, SPA’s advantage is most pronounced on hard localization cases: there are modules where the baseline remains at 0% success while SPA variants achieve non-zero pass@1, indicating that SPA can “unstick” the model in situations where the original prompt repeatedly fails, even though the surrounding framework is unchanged.
>
> **Why we did not further ablate every framework component.**
> We fully agree that ablations over individual components (e.g., removing hierarchical matching or changing selection strategies) would be interesting for system designers. However, there is a large combinatorial space of pipeline variations. Exploring it in depth would substantially broaden the scope and distract from our central question:
>
> > Does stochastic prompt perturbation provide a robust, input-level exploration mechanism under a fixed debugging workflow?
>
> To keep the scope focused, we therefore:
>
> - Use SPARC-Debugger as a single, shared backbone across all methods,
> - Focus on comparisons where SPA is the only changing factor, and
> - Conduct detailed ablations *within SPA itself* (noise type, length, position, temperature) in Section 5.3.
>
> We will clarify this design choice in the revision, explicitly stating that SPARC-Debugger is primarily used as a platform for fair, equal-budget evaluation of SPA versus the original prompt.
>
> ---

---

> ### Author Response · Authors · 2025-11-25
>
> ### (3) How we choose perturbation strategies
>
> > **Reviewer’s concern (Weakness 3; Q3).**
> > “The detail implementation of choosing perturbation strategy and generate perturbation is also not fully illustrated.”
>
> Our strategy selection is empirical and task-driven, guided by the ablations in Sections 5.2–5.3. There are two main regimes.
>
> ### (3a) For Verilog debugging (CirFix)
>
> For debugging, the main challenge is escaping persistent failure modes caused by wrong error localization. In this setting, our experiments consistently show that:
>
> - Among all tested variants, **PRN+End** provides the most robust gains across models.
> - Appending syntactic noise at the end preserves the structure of the compiler message and the buggy code, while still perturbing the surrounding context enough to change the model’s attention and reasoning path.
> - Moderate noise length (around 30% of the prompt) gives the best trade-off: too short has little effect; too long starts to dilute the useful signal.
>
> Based on these observations, we adopt **PRN+End with a moderate noise length** as the default SPA configuration for CirFix debugging.
>
> ### (3b) For Verilog generation (VerilogEval)
>
> For generation, the situation is different:
>
> - The baseline (no-perturbation) performance is already strong.
> - SPA is mainly used as a “portfolio” mechanism to diversify first-pass candidates, rather than to rescue completely failing prompts.
>
> Our ablations show a model- and task-dependent pattern:
>
> - For **Qwen3-32B**, **PRN+Front** consistently works best on both Code Completion and Spec-to-RTL tasks. Injecting syntactic noise at the front slightly perturbs how the model interprets the high-level instruction, encouraging diverse but still well-formed implementations.
> - For **GPT-4o-mini**, **SRN+End** is the strongest variant in our experiments: structured syntactic noise appended as comments at the end expands the solution space without interfering with the spec at the beginning of the prompt.
>
> In practice, for generation tasks we therefore pick the best-performing syntactic variant from these ablations (PRN+Front or SRN+End, depending on the model) and keep it fixed.
>
> ### (3c) Why we avoid KIN in practice
>
> Our experiments also show that Knowledge-Influenced Noise (KIN) consistently underperforms PRN/SRN across both debugging and generation: injecting task-irrelevant but semantically meaningful tokens tends to distract the model and degrade output quality. As a consequence, KIN is not part of our recommended configurations; we focus on syntactic perturbations that change internal attention patterns without introducing misleading semantics.
>
> ### (3d) Practical use: combining multiple strategies
>
> Finally, we emphasize that in real debugging or generation workflows, our method does not require committing to a single, fixed perturbation strategy. Noise generation is purely prompt-side and computationally cheap, so practitioners can easily allocate a small query budget to a portfolio of SPA variants (e.g., combining PRN+End and PRN+Front, or mixing PRN and SRN) within the same overall number of LLM calls. In the paper, we choose one strong configuration per regime (PRN+End for debugging, PRN+Front / SRN+End for generation) for clarity of presentation and controlled evaluation, but the framework is designed to support such combinations in practice.

---

### Note · Program_Chairs · 2026-01-17
**Submission Desk Rejected by Program Chairs**

The following references in this submission do not refer to real documents and/or have major errors in bibliographic information:

 Md Arafatur Rahman, Chanchal K Roy, and Kevin A Roy. A survey on software debugging. In 2015 IEEE 22nd International Conference on Software Analysis, Evolution, and Reengineering (SANER), pp. 463-467. IEEE, 2015. While broadly about software debugging, it highlights the generally labor-intensive nature of debugging complex systems, a characteristic applicable to hardware description languages like Verilog. More specific hardware debugging surveys also echo this sentiment regarding effort.
Yiorgos Makris, Emad Oraizi, and Nicola Nicolici. Defect and fault tolerance in VLSI systems: A survey of methodologies and techniques. ACM Computing Surveys (CSUR), 38(3):8-es, 2006.